# Objecthood, Agency and Mutualism in Valenced Farm Animal Environments

**DOI:** 10.3390/ani8040050

**Published:** 2018-04-03

**Authors:** Ian G. Colditz

**Affiliations:** CSIRO Agriculture and Food, Armidale 2350, Australia; ian.colditz@csiro.au; Tel.: +61-267-761-460

**Keywords:** agency, homeostasis, allostasis, mutualism, resilience, immune function, flight speed

## Abstract

**Simple Summary:**

Selective breeding and intensive management of farm animals have increased their productivity. At the same time, the ability of the animal to cope with challenges from its environment has decreased. The manner in which the animal interacts with its environment influences its ability to cope. In recent models of brain function, neural tissue is understood to predict the sensations that arise from behavioural activities. The brain also predicts sensory input from physiological activities and from at least some activities of the immune system. An ability to predict and control sensory inputs provides the animal with agency, while a continuing discrepancy between predicted and actual sensory inputs leads to stress and negative emotional states. Through these processes, aspects of the environment acquire a negative or positive character: that is the environment becomes valenced. Predicting and controlling its environment gives the animal agency. Mutualism may represent a further step towards closer synchronisation between animal and environment. A better description of the environment from the animal’s perspective could improve our understanding of how cognition, behaviour, temperament, immune functions and metabolic activities are linked, and improve strategies to enhance farm animal welfare.

**Abstract:**

Genetic selection of farm animals for productivity, and intensification of farming practices have yielded substantial improvements in efficiency; however, the capacity of animals to cope with environmental challenges has diminished. Understanding how the animal and environment interact is central to efforts to improve the health, fitness, and welfare of animals through breeding and management strategies. The review examines aspects of the environment that are sensed by the animal. The predictive brain model of sensory perception and motor action (the Bayesian brain model) and its recent extension to account for anticipatory, predictive control of physiological activities is described. Afferent sensory input from the immune system, and induction of predictive immune functions by the efferent nervous system are also in accord with the Bayesian brain model. In this model, expected sensory input (from external, physiological and immunological environments) is reconciled with actual sensory input through behavioural, physiological and immune actions, and through updating future expectations. Sustained discrepancy between expected and actual sensory inputs occurs when environmental encounters cannot be predicted and controlled resulting in stress and negative affective states. Through these processes, from the animal’s perspective, aspects of the environment acquire a negative or positive character: that is the environment becomes valenced. In a homeostatic manner, affective experience guides the animal towards synchronisation and a greater degree of mutualism with its environment. A better understanding of the dynamic among environmental valence, animal affect and mutualism may provide a better understanding of genetic and phenotypic links between temperament, immune function, metabolic performance, affective state, and resilience in farm animals, and provide further opportunities to improve their welfare.

## 1. Introduction

The application of quantitative methods to the selective breeding of farm animals and the intensification of animal management in the second half of the 20th century have led to impressive improvements in productivity. At the same time as meat, milk, fibre, and egg production have increased, the animal’s capacity to stay healthy and fertile in a diversity of environments has diminished [1]. The causes for this disjunction between productivity and resilience to environmental challenges are drawing increasing attention from genetic, physiological and behavioural research. Competition at the genetic level between production and fitness (functional) traits has been examined for instance through resource allocation theory [2], while balanced genetic selection for production and fitness is increasingly being implemented [3,4,5]. In addition, studies on the impacts of housing and management practices on behaviour and emotional (affective) states are underpinning improvements in animal welfare [6]. Physiological and behavioural studies are increasingly identifying links between temperament, stress, resistance to infection and growth [7,8,9].

The relationship between the animal and its environment is central to all these aspects of farm animal biology. Is the environment a concrete, invariant, objective reality that shapes the animal, or do animal and environment engage in a more interdependent manner? This brief review examines animal environment interaction and its relationship to neural, behavioural, physiological and immune functions of the animal. As the central integrative modulator of these processes, it is suggested that affective state provides a barometer of mutualism in the dynamic of animal environment engagement.

## 2. The Animal’s Relationship with Its External Environment

Does the environment exert a mechanical influence on the animal in its everyday life? This materialist conception of an instrumental environment shaping the animal was challenged in 1909 by the Estonian biologist von Uexküll who proposed that the animal is immersed in its surrounding world. Writing in German, he termed the surrounding world “Umwelt” and proposed that the animal and its environment constitute a functional interacting unit, or function circle [10,11]. Von Uexküll proposed that for each species the Umwelt is structured in signs or marks that the animal senses. In von Uexküll’s view the environment for each species is subjective and unique. 

In 1977, the American psychologist JJ Gibson introduced the concept of “affordances” to describe the resources available to the animal in its environment [12,13]. Gibson proposed that “[t]he affordances of the environment are what it offers the animal, what it *provides* or *furnishes*, either for good or ill [emphasis in original]” (p. 127). In Gibson’s view, affordances are features of the environment that provide opportunities for action. Affordances can be surfaces to stand on, to lie on or to rub against, objects to chew, air to breathe, and so on. For a goat climbing on a cliff face to access plants and salt, the crevices and ledges are affordances, but for cattle (for example) these same ledges may provide no affordance [14]. Opportunities for action provided by the environment can change as the animal develops through its life history [15]. Two species in one location may interact with differing sets of affordances that for each species constitutes its own niche. Within a population, individuals may differ in the affordances they perceive [16]. In contrast to von Uexküll’s Umwelt as a subjective environment, Gibson and his colleagues considered that affordances have an objective character [17,18]. Despite this difference, both accounts portray a dynamic, hand-in-glove fit between the animal and its environment. Patten [19] further built on these two concepts to propose that the organism and the environment do not constitute a dualism bound together through transaction, but rather combine to create an emergent property in nature as a complementary interactional unity. From these and other lines of enquiry (e.g., [20,21,22]) has developed the concept that the organism and its environment do not interact in a dualism as separate entities but rather combine together in mutualism as a dynamic process.

## 3. Perception of the External Environment

Recognition of animal/environment as a dynamic system starts from a consideration of the animal as a whole functional entity. Asking how the animal’s sensory systems enable it to engage with its environment begins to break the animal down into functions or activities, and leads to questions of how senses enable neural control of behaviour and physiology. The brain is physically isolated from the external world yet requires information to regulate engagement of the animal with its surroundings. The senses of sight, hearing, touch, taste and smell as well as the senses of position and motion of the body send afferent signals to the brain reporting the animal’s interaction with the external environment [23]. The brain can modify the information it receives by sending efferent signals to muscles that initiate motor actions that change the animal’s engagement with its environment. These sensorimotor actions can lead to a new set of stimuli invoking the next train of afferent sensory signals sent to the brain [24].

Recent models of neural function suggest that rather than operating as a stimulus response network, the brain generates expectations from prior experience of the train of sensory input noted above that it is about to receive. Discrepancy between expected and actual sensory input can lead both to sensorimotor actions and to a change in subsequent expectations [25,26,27]. Here “expectations” are a current state of neurophysiological activity and “discrepancies” are discordances in neurotransmitter signals and action potentials between afferent and efferent neural tissues. The outcome of these neurophysiological activities occurring within the hierarchical architecture of the brain is a self-organising system that maintains its internal integrity by modifying its engagement with the external environment [26]. The analogy most commonly used to describe this process is the statistical procedure of approximate Bayesian inference [27]. Thus the brain is described as a Bayesian brain using predictive coding to generate active inferences about the environmental causes of sensations and perceptions [28]. It is important to recognize that the “Bayesian brain” analogy does not imply that an algorithm-based statistical computation underpins perception and sensori-motor actions [22,29]. Furthermore, the Bayesian brain analogy does not displace decades of research on the role of neurophysiological and neurohormonal networks in controlling physiological and behavioural states like hunger, appetite and exploration, but rather provides an overarching and unifying explanatory account of how predictive self-organising and self-preserving physiological and behavioural activities arise as discussed further below. For detailed accounts see [25,26,30,31].

## 4. The Animal’s Relationship with Its Internal Environment

As well as controlling engagement with the external environment, the brain maintains the internal environment of the animal in a physiological state conducive to surviving, thriving, and reproducing [25,32]. Sensory inputs from peripheral tissues relay information from physiological, vascular, metabolic, neuroendocrine, autonomic and visceral functions to produce the sense termed interoception [23,33,34]. Anatomical and neurophysiological evidence indicates that the internal environment is regulated through predictive and anticipatory actions that are again analogous to approximate Bayesian inference [25,30,31,35,36]. These accounts at the neural level are in accord with over a century of empirical evidence of predictive, anticipatory actions maintaining homeostasis [37,38,39]. In 1988, Sterling and Eyer [40] termed this well recognised pattern of predictive homeostatic regulation allostasis meaning “stability through change” [41].

An important feature of interoception is its influence on the affective state of the animal [42,43,44]. While the influence of the external environment on affective state has long been recognised [44,45,46], the more recent realisation that interoceptive perception of metabolic state influences affect [33] has important implications for animal welfare. For instance, there may be an important influence on affective state of extended periods of negative energy balance during restriction of feed intake to manage reproductive performance (e.g., broiler hens), during drought, or during early lactation in high performing dairy cows. Thus, a distinction may exist between interoceptive perception of metabolic state and central perception of unfulfilled drives (e.g., hunger) as sources of negative affect. 

## 5. The Animal’s Relationship with the Immunological Environment

The immune system of vertebrates recognizes short sequences and three dimensional conformations of molecules through two sets of receptors. The first is a germ line encoded set that binds predominantly with molecular structures common amongst microorganisms. These receptors provide innate immunity. A second set of receptors are generated throughout the life of the animal by somatic hypermutation of a small number of genes and provide adaptive immunity. A vast network of messengers produced within the immune system regulates its activity [47]. In addition, the brain has extensive afferent and efferent innervation of the immune system as well as bi-directional hormone and cytokine communication via blood [48,49,50].

Two important questions are firstly whether immune activity as a sensory input to neural tissue can be linked to anticipatory behavioural and physiological functions, and secondly whether the brain can exert anticipatory and predictive control over activities of the immune system. The role of the immune system as an afferent sensory arm of the brain is well recognised. For instance, immune activation by a wide variety of immune stimuli induce sickness behaviours that modify foraging and rest [51,52], a negative affective state [53] and alteration of metabolic activities [54,55]. These physiological and behavioural responses are predictive and anticipatory in that they can be initiated early in a disease setting before metabolic demands of infection are well advanced and before pathological damage to tissues has been initiated by the infectious agent. These changes are anticipatory in the same sense as the increase in heart rate, blood flow to muscle and glucose mobilisation that occur before exercise are anticipatory of the metabolic demands of muscle during exercise. Thus, the afferent sensory inputs from the immune system to the brain conform with the Bayesian model of neural control of physiology and behaviour.

The other arm of the Bayesian model is initiation of predictive actions. This is the second question. Can the brain exert predictive control over immune functions? Indeed a large body of evidence indicates that a wide range of immune functions can be entrained by classical Pavlovian conditioning to non-immunological cues such as ingestion of a sucrose syrup in rats [49,56,57]. In this setting, a non-immunological stimulus (such as sucrose syrup) evokes activities of the immune system such as production of antibodies, activation of T lymphocytes, release of cytokines, initiation of fever and activation of the acute phase response [57]. Other predictive actions of the brain on immune function seem likely. Pavlovian conditioning of immune responses provides evidence that the neuro-immune dialogue exhibits a pattern of predictive control of immune function as seen for the sensori-motor system and the interoceptive-homeostatic system of physiological activity.

## 6. Environmental Boundaries

In the formulation given above of animal/external environment as dynamic process, there is no discrete boundary between the animal and the environment that determines where the animal ends and the external environment begins [20]. This is most easily understood through the examples of tools, birds’ nests and exoskeletons (see [20,58] for detailed accounts of these and other examples of extended phenotype and extended mind). A practical example in farm animals is provided by the heritability of social effects on production traits such as growth rate, feed intake, back fat thickness and muscle depth in pigs [59]. While the genes influencing these traits lie within the animal, a component of the heritable genetic variation in these traits lies within the social environment outside the individual animal. 

As the internal physiological environment is contained within the animal, a consideration of its boundary with the animal need not concern us in detail here; however, the nature of the immunological boundary poses a more interesting problem. The boundary between the animal and the environment is described by immunologists to arise from a distinction between self (molecular structures of the tissues of the host animal) and non-self (the molecular structures of entities that are foreign to the animal) [60]. Further, the capacity for the immune system to distinguish between self and non-self is described as immune agency. Here I follow Tauber’s account of development of the concept of immune agency [61]. The concept grew out of Burnett’s clonal selection theory, which is predicated on the notion that new antigen receptors generated by somatic hypermutation must be negatively selected to avoid reactivity against the tissues of the host as well as positively selected for affinity to foreign (non-self) antigens. From this theory the idea emerged that the host has a fixed antigenic identity that is dictated by its genome, and that the immune system acts as defender and interrogator of that identity [61]. However there is a growing realization that the vertebrate host is comprised of a community of symbiotic organisms that changes during juvenile development and over subsequent time periods [62]. This community of organisms, which together with the host animal is described as a holobiont, provides a conceptual challenge to the notion that the host animal possesses a unique immunological self. As well as the presence of beneficial organisms on the skin and mucosal surfaces of the animal, there is ingestion in the diet of small membrane bound particles termed exosomes that contain proteins, RNA and DNA from plants, animals and microbes [63]. Some of these exosomes cross the gut wall and are taken into the cells and tissues of the host where they can express biological functions native to their plant, animal or microbial source. Thus, there is no distinct boundary between the molecular identity of the host animal and the microbial organisms and dietary constituents it engages with. This leads to the view that the immune system mediates a dialogue about molecular structures between the holobiont and its environment, rather than acts as defender of a genomically prescribed immunological self [61]. This understanding is described by immunologists as an ecological concept of immune function (eco-immunology) [64]. Tauber notes that “[i]n this mutualist setting, immune identity is dynamic and adjusts to the needs and opportunities offered by the environment” (p. 221). Thus, the immune system does not operate to guard a genomically defined immunological self, but rather “organismic identity emerges in dynamic encounters with the world (both within the body of the animal and beyond) in a world fraught with various friend and foe relationships” (p. 222).

Thus, whether examining the animal as a behavioural agent sensing sounds, smells, light, movement and pressure, or as an immunological agent sensing molecular structures, there is no clear boundary between the organism and the environment. Within the environment of affordances and the environment of molecular structures, the organism and environment combine as a dynamic process.

## 7. Equilibrium within the Environment/Animal Process

The Bayesian account sees behavioural, physiological and immune functions as processes in which expectations are continually updated to better match environmental conditions. Expectations are generated endogenously from prior experience and from drives and needs arising spontaneously from neural activities. Yet environmental conditions are rarely static and the capacity to reconcile expectations with actual afferent sensory inputs can be limited by the ability of the animal to predict change in, or to exert influence over its external, physiological and immune environments. An inability to predict and control environments results in changes that are recognised as stress [26,65,66]. In each domain, failure to reconcile expectation with sensation leads to negative affective experience that influences perceptions and actions in the other domains. Thus, the Bayesian model provides an integrative account of the interdependencies between behaviour, physiology and immune processes through a shared influence on the subjective experience of affect. Nonetheless, it is noteworthy that models of this kind do not necessarily predict the mechanistic detail of biological functions at the molecular and cellular detail. For instance, the concepts of homeostasis and allostasis do not predict the role that biochemical mediators, receptors, cell responses and behavioural actions play in regulation of any specific physiological variable. Despite this limitation, the concepts of homeostasis and allostasis play a foundational role in understanding physiology. In this respect then, the inability of the Bayesian model to predict molecular mechanisms is not necessarily a short coming.

Within the dynamic of animal/environment interaction, the capacity of the animal to balance sensations with expectations is not necessarily stable over time. When actions are unable to provide prediction and control, the environment can be considered to exert an instrumentality that shapes the actions of the animal. As an object of environmental effects, the animal is in a state of objecthood. When the animal can command its interactions with the environment through prediction and control it exerts agency [27,67]. As recognised in the progressive severity of stress [62], environmental instrumentality and animal agency exhibit a reciprocal dynamic [26]. This formulation helps resolve a limitation of the Bayesian model, which sees the circle of interaction between animal and environment as starting with an animal’s expectations. In the Bayesian model, action does not depend upon an initiating environmental stimulus leading to an animal response, rather expectation precedes sensation. Yet novel objects and unprecedented sequences of events do occur in the animal’s environment and lead to a causal sequence where actions follow after environmental change. Thus, in the reciprocal dynamic model, the start of the circle of interaction is seen to fluctuate between environment and animal. 

Yet acting as agent may not itself represent the optimal outcome for the animal of the animal environment interaction. As noted above, the dynamic of interaction generates affective experience. From this realisation developed the concept that, from the animal’s perspective, the environment is valenced and provides a landscape of affective experience [22,44,68,69]. The concept of a valenced landscape is also intrinsic to Gibson’s affordances (for “good or ill”) and Tauber’s molecular dialogue (“in a world fraught with various friend and foe relationships”). In a homeostatic manner, both positive and negative valence may guide the animal towards a state of greater synchronisation with its environment leading to a progression from agency to a more felicitous state of mutualism [46,68]. In this process lies a progression from “me” (object), to “I” (subject), to “one” (impersonal pronoun, connoting mutual participant). The progression can be illustrated with the metaphor of “me” a seasick sailor tossed about in a boat bobbing haphazardly on a choppy sea becoming “I”, the captain, sail set and hand on tiller steadying the boat, becoming “one”, the tillerman, working with the rise of the swell, the run of the tide and the luff of the sail to steer a course across an unresting ocean. Whereas agency is the ability to proactively influence and control the environment, mutualism may be the ability to synchronise actions with the anticipated valence of imminent environmental changes thereby minimising deviations in the affective state. 

## 8. Farm Animal Environments

Farm animals are managed in a diversity of environments extending from naturalistic rangeland pastures to intensive housing in close confinement with conspecifics. Availability of environmental resources and freedom to fulfil needs and drives are recognized as fundamental requirements for good animal welfare [70]. The need to reduce the objecthood of animals by facilitating the development of their agency is an objective of much current research [71,72]. Is mutuality achievable in farming systems or is agency sufficient? The central role of affective state as modulator of cognitive, behavioural, physiological and immune processes, and the potential for the dynamics of affective state to be a monitor of mutuality as suggested above, highlight the importance of measures of affective state [73]. Substantial progress in this endeavour has been made through behavioural methods including cognitive bias, attention bias, startle tests, and acoustic analysis of vocalisations [6,74,75,76], and through physiological measures including heart rate and heart rate variability [77,78]. Negative affective states associated with anxiety and with husbandry procedures have proved easier to measure than positive affective states. An example of a reduction of negative affect that may be suggestive of mutualism is provided by the “call feeder” system in pigs. In this system, each sow learns a personal auditory cue that signals a period of uncontested access to a communal feeder [79]. The personal signal structures the environment to provide each individual with a unique opportunity to control social conflicts at the feed station, which leads to improved affective experience [80,81]. A substantial dilemma exists when measuring affective state in determining where along a continuum from negative to positive, less negative affect represent progression past a neutral settling point (core affect) into a state of positivity.

## 9. Final Remarks

From many lines of enquiry comes a view that the animal needs a cognitive, affective, physical and immunological grasp on its environment in order to survive, thrive and reproduce. While substantial progress has been made, an improved understanding of aspects that enable farm animals to predict and control their environment has the potential to further improve definition of traits such as temperament and resilience, and to further improve design of the farm animal environment. In outbred populations like farm animals, individuals differ in their response to conditions and challenges that to the scientist’s eye may appear identical for each animal [9,82,83]. Both the genetic and the environmental causes of these differences are of interest. Genetic selection for favourable behavioural response patterns can provide a robust means for improving health and production [84]. For instance, flight speed in cattle is favourably correlated with health outcomes, growth rates and meat quality [85,86,87,88]. Flight speed is measured as cattle voluntarily cross a defined distance (typically 1.7 m) when the crush (chute) they are held in is opened by a stockperson. Despite the empirical utility of flight speed and many other behavioural tests as predictors of production performance, aspects of the environment that influence animal activity in these tests are typically poorly understood. Here the challenge is to move from descriptions of the environmental conditions that constitute the test in terms like distance, duration, light intensity, facility design, presence of conspecifics, and so on to a description of the features the animal perceives and engages with in a manner that influences its activity and affective experience. The need for an animal’s view of the environment has been recognised for many years [6,89,90,91]. Continuing to improve our understanding of the animal’s perspective should enable a better characterisation of the environment and a better understanding of how environmental features contribute to genetic and phenotypic links between temperament, immune function, metabolic performance, affective state, resilience, and ultimately welfare. Restructuring the cognitive, affective, and physical aspects of the production environment to facilitate “control through prediction” as a steppingstone towards “mutualism through synchronisation” should yield benefits across all dimensions of biological function.

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
