# Peer review of "Objecthood, Agency and Mutualism in Valenced Farm Animal Environments"

_animals, 2018, doi:10.3390/ani8040050_

Round 1

Reviewer 1 Report

p.p1 {margin: 0.0px 0.0px 0.0px 0.0px; font: 11.0px 'Helvetica Neue'; color: #000000; -webkit-text-stroke: #000000} p.p2 {margin: 0.0px 0.0px 0.0px 0.0px; font: 11.0px 'Helvetica Neue'; color: #000000; -webkit-text-stroke: #000000; min-height: 12.0px} li.li1 {margin: 0.0px 0.0px 0.0px 0.0px; font: 11.0px 'Helvetica Neue'; color: #000000; -webkit-text-stroke: #000000} span.s1 {font-kerning: none} ol.ol1 {list-style-type: decimal}

This review on relationship between animals and their environment is difficult to read. It was hard to review. This is, in part, because it uses a theoretical language that introduces terms that are not are in common use in this area of science without appropriate explanation and definition. There is no “easing” the reader in to these conceptual arguments. This will mean that the uninitiated reader will struggle, to the point where few will likely continue to read the paper. Then the impact will be lost. For example, a new graduate student will struggle with the simple summary (which is far from simple), will hit hurdles in the abstract confronted without explanation with “mutualism”, “umwelt”, “affordances”, “agency” and so on. Unfortunately, I very much doubt that this manuscript, if published, would encourage a reader to explore these concepts in more detail, as the author states.

In addition to the language and style, this paper lacks clear direction and an objective. There is certainly no clear hypothesis. The objective of encouraging readers to explore particular concepts for themselves is not sufficient. The author should be providing assistance in this. There is an opportunity to provide a clear and critical review of the literature with solid interpretations from the author but this is not what has been presented. This is a somewhat selective historical account of some literature that has not been adequately synthesised and interpreted by the author. What is the take-home message?

In some ways, section 7 of the paper, limitations, underscores many of the issues with this manuscript. The conclusion is hardly that and the final paragraph of the manuscript is far from an acceptable and useful take-home conclusion.

Two specific comments:

Allostasis had not been adequately, or appropriately, discussed. Given that this is a review, the work of McEwen and colleagues, at the very should have been included.

Page 2, line 59, “…we think…”, the reader will not appreciate being told what to think.

Author Response

Thank you for your comments. My replies are in red text.

This review on relationship between animals and their environment is difficult to read. It was hard to review. This is, in part, because it uses a theoretical language that introduces terms that are not are in common use in this area of science without appropriate explanation and definition. There is no “easing” the reader in to these conceptual arguments. This will mean that the uninitiated reader will struggle, to the point where few will likely continue to read the paper. Then the impact will be lost. For example, a new graduate student will struggle with the simple summary (which is far from simple), will hit hurdles in the abstract confronted without explanation with “mutualism”, “umwelt”, “affordances”, “agency” and so on. Unfortunately, I very much doubt that this manuscript, if published, would encourage a reader to explore these concepts in more detail, as the author states.

The paper and abstracts have been largely re-written and shortened in accord with this comment and comments of other reviewers. A more straight forward explanatory style with less terminology is now used.

In addition to the language and style, this paper lacks clear direction and an objective. There is certainly no clear hypothesis. The objective of encouraging readers to explore particular concepts for themselves is not sufficient. The author should be providing assistance in this. There is an opportunity to provide a clear and critical review of the literature with solid interpretations from the author but this is not what has been presented. This is a somewhat selective historical account of some literature that has not been adequately synthesised and interpreted by the author. What is the take-home message?

The objective of the paper is now tied to issues in farm animal biology as explained in a new introduction, and returned to in the “Final remarks” section

In some ways, section 7 of the paper, limitations, underscores many of the issues with this manuscript. The conclusion is hardly that and the final paragraph of the manuscript is far from an acceptable and useful take-home conclusion.

With the change in style and scope of the paper, Section 7 is now deleted, and the final section of the paper is focussed on the issues in farm animal biology outlined in the introduction

Two specific comments:

Allostasis had not been adequately, or appropriately, discussed. Given that this is a review, the work of McEwen and colleagues, at the very should have been included.

McEwen and colleagues work is now addressed. In particular, their recent account of the neural basis of stress and allostasis is described in detail in Section 6.

Page 2, line 59, “…we think…”, the reader will not appreciate being told what to think.

This section has been re-written (lines 45-66)

Reviewer 2 Report

It is a nice brief scientific review of animals and related environments with a wide range of their relationship concepts.

Although it is superficial it has reached the author’s aim: a brief introduction of ecological models of the animal-environmental interaction. As the author indicated: The information will enhance the understanding of animal behavioral, physiological and immune functions within their environments, i.e., animal-environmental interaction, and provide a sight for better understanding animal capability to adapt to their environments, resulting in developing better management to improve animal health and welfare.

Author Response

Thank you for your comments. My replies are entered with red highlights

It is a nice brief scientific review of animals and related environments with a wide range of their relationship concepts.

Although it is superficial it has reached the author’s aim: a brief introduction of ecological models of the animal-environmental interaction. As the author indicated: The information will enhance the understanding of animal behavioral, physiological and immune functions within their environments, i.e., animal-environmental interaction, and provide a sight for better understanding animal capability to adapt to their environments, resulting in developing better management to improve animal health and welfare.

The focus on farm animal biology is now increased to improve relevance to the readers of Animals.

Reviewer 3 Report

This is a thoughtful and well-informed article, and the issue of mutualism is close to my own heart! I am grateful, for example, to find out about Walsh’s book, which sounds great (Walsh, D.M. Organisms, agency, and evolution; Cambridge University Press: 2015) .

BUT - and I don’t think it has to be a very big BUT - this article is too eclectic. A main focus is the work of Gibson. Gibson was not very consistent theoretically. But when he was being most radical, I don’t think his approach can be so easily assimilated to the two main claims of this paper.

1.    von Uexküll, and Patten (whose work I read when it first appeared) present an overly subjectivised notion of the “environment” that Gibson, with his concept of affordances, was trying to undermine, without ending up, instead, with objectivism.

2.    Gibson’s concept of information was an attempt to counter the kind of cognitive/computational concepts invoked in this paper (inferences, Bayesian computation). As Gibson put it, from a mathematical viewpoint life might seem complicated, but from a biological one it may well be simple. (The best papers on this are by my friend Sverker Runeson, see below)

So, in short, I like where this paper is heading, but it needs to be much more alert to the fact it is trying to reconcile apparently incompatible theoretical approaches. Done self-consciously, this could be productive.

Homework:

1.    The very best recent paper on mutualism is: Daniel K. Palmer ON THE ORGANISM–ENVIRONMENT DISTINCTION IN PSYCHOLOGY Behavior and Philosophy, 32, 317-347 (2004),

2.    An important figure in recent evolutionary biology arguing for mutualism is Lewontin: e.g. R. C. Lewontin, SOCIOBIOLOGY AS AN ADAPTATIONIST PROGRAM. Behavioral Science, Volume 24, 1979.  The only big problem is that he misrepresents Darwin as a Cartesian: See Costall, A.  (2004).  From Darwin to Watson (and Cognitivism) and back again: the principle of animal-environment mutuality.  Behavior & Philosophy, 32, 179-195.

3.    Runeson, S. (1980). There is more to psychological meaningfulness than computation and representation. Behavioral and Brain Sciences, 3, 399-400. Runeson, S. (1977). On the possibility of smart perceptual mechanisms. Scandinavian Journal of Psychology, 18(1), 172-179.

Author Response

Thank you for your comments, My replies are entered in red highlight

This is a thoughtful and well-informed article, and the issue of mutualism is close to my own heart! I am grateful, for example, to find out about Walsh’s book, which sounds great (Walsh, D.M. Organisms, agency, and evolution; Cambridge University Press: 2015) .

BUT - and I don’t think it has to be a very big BUT - this article is too eclectic. A main focus is the work of Gibson. Gibson was not very consistent theoretically. But when he was being most radical, I don’t think his approach can be so easily assimilated to the two main claims of this paper.

In the re-working of the paper, the focus on Gibson is greatly reduced (but not removed) and emphasis placed on more recent work such as Palmer, Costall (thank you very much for these and other references) and Bruinesberg and colleagues. In this manner, less emphasis is placed on a link between Gibson and mutualism. Indeed mutualism itself is not a strong focus of the revised paper. This is not from a loss of heart for its importance on my part, but rather due to the greater emphasis placed on the work of Peters, McEwen and Friston on stress to describe the link between predictive control and agency on the one hand ,and lack of control and stress on the other. Their account also provides a practical and detailed implementation of the Bayesian brain model and so, I think, helps make the predictive brain concept more believable to readers not familiar from outside the field. Stress helps illustrate the dynamic character of agency. In terms of mutualism, I think the account provided by Bruinesberg and colleagues of environmental grip (referenced in the paper) is particularly valuable in the way it links neurobiology with an ecological concept of how the animal engages with its environment, and so may be of interest to you.

1.    von Uexküll, and Patten (whose work I read when it first appeared) present an overly subjectivised notion of the “environment” that Gibson, with his concept of affordances, was trying to undermine, without ending up, instead, with objectivism.

2.    Gibson’s concept of information was an attempt to counter the kind of cognitive/computational concepts invoked in this paper (inferences, Bayesian computation). As Gibson put it, from a mathematical viewpoint life might seem complicated, but from a biological one it may well be simple. (The best papers on this are by my friend Sverker Runeson, see below)

Thank you for these two comments. Within the revised paper there is less emphasis now on subjectivism / objectivism, and perhaps a little more on the Bayesian brain model. However, drawing on Runeson’s work, I strongly emphasize that the Bayesian brain model provides an analogy to describe the outcome of the continually updating loop between action and sensation, rather describes a process of algorithmic calculation of prior probabilities, and so on. Indeed, I think the computational aspect of the Bayesian brain model is greatly over emphasized in most accounts. The manner in which activities of action potentials and neuro transmitters operating within the hierarchical architecture of the cortex can lead to “expectations” and “error prediction” etc is being realised in impressive detail in some studies, especially of vision. That these outcomes look like the product of approximate Bayesian inference is just a hand analogy to describe the process, I think.  Also, I think predictive brain control of homeostasis is over-emphasized in most accounts, and I try to make this point also. It is the predictive dimension of control that is provided by the Bayesian brain, not control of the whole homeostatic/ allostatic process, not all or which is predictive.

The subjectivism / objectivism issue is very important. If I may digress for a moment, I’ve been think that exploring the issue as you suggest might well provide a way to approach the problem of reconciling in narrative form the difference between thermodynamic information, as understood in statistical mechanics and biological information, as understood in Information Theory.  I’ve removed “information” from the revised manuscript, so addressing the issue is now very much outside the scope of the paper. My thought here is that, notwithstanding the impressive detail arising from psychophysics type analyses of the invariant information that give affordances their grippability (in Bruinesberg et al’s terminology), biological information within Patten’s environ does indeed have a degree of subjectivity. It is the dependence in Information Theory (and in Jablonka and Lamb’s account of information in Evolution in 4 Dimensions, and Irun Cohen’s account of information in Tending Adam’s garden, and Peirce’s Theory of Signs, and so on) for there to be a “receiver” that transforms thermodynamic information into biological information. For Shannon, information is “a reduction in uncertainty”. Separate from the ability to describe the statistical dimension of uncertainty (Shannon entropy) in precise mathematical terms is the realisation that uncertainty is conditional on the need for a receiver to act (or have the potential to act). So if Shannon’s information is a “reduction in uncertainty” then when a receiver reduces uncertainty they are doing work to reduce what Friston calls information free energy. This is the equivalent in statistical mechanics to a machine using Gibbs free energy to do (thermodynamic) work. Point here if my line of thinking is on target is that thermodynamic information becomes biological information by the need for a receiver to reduce uncertainty. Subjectivity lies in the fact uncertainty is a state that is relative to the ability of the organism to sense and reduce it. That state is separate from the statistical account provided by measuring Shannon Entropy. These discussions can become circular, but the receiver here in Shanon’s information theory is in effect an organism, or tool of an organism. So, Maxwell’s demon is Palmer’s bioprocess.  Stuff in the environment becomes information when a receiver has uncertainty and a need to act (or an interest in the potential to act). For instance electromagnetic radiation from the Big Band becomes information when humans are uncertain of the age of the universe and try to answer the question, or at least reduce the uncertainty. In this formulation, Information Theory is an enactivism theory. But this, if there is an account to be made, is beyond the scope of the paper.    

So, in short, I like where this paper is heading, but it needs to be much more alert to the fact it is trying to reconcile apparently incompatible theoretical approaches. Done self-consciously, this could be productive.

Homework:

1.    The very best recent paper on mutualism is: Daniel K. Palmer ON THE ORGANISM–ENVIRONMENT DISTINCTION IN PSYCHOLOGY Behavior and Philosophy, 32, 317-347 (2004),

2.    An important figure in recent evolutionary biology arguing for mutualism is Lewontin: e.g. R. C. Lewontin, SOCIOBIOLOGY AS AN ADAPTATIONIST PROGRAM. Behavioral Science, Volume 24, 1979.  The only big problem is that he misrepresents Darwin as a Cartesian: See Costall, A.  (2004).  From Darwin to Watson (and Cognitivism) and back again: the principle of animal-environment mutuality.  Behavior & Philosophy, 32, 179-195.

3.    Runeson, S. (1980). There is more to psychological meaningfulness than computation and representation. Behavioral and Brain Sciences, 3, 399-400. Runeson, S. (1977). On the possibility of smart perceptual mechanisms. Scandinavian Journal of Psychology, 18(1), 172-179.

Thank you very much for the homework which was very interesting and has influenced the modified version of the paper.

Reviewer 4 Report

This is a paper by a philosopher. Whilst philosophers have much to offer in animal-related studies, I do not find this approach useful. Unless the nature of interactions are specified I think the discussion too superficial to be meaningful. hence I do not think the paper acceptable for Animals.

Author Response

This is a paper by a philosopher. Whilst philosophers have much to offer in animal-related studies, I do not find this approach useful. Unless the nature of interactions are specified I think the discussion too superficial to be meaningful. hence I do not think the paper acceptable for Animals

Thank you for your comments. My comments are entered with red highlight

My interest in the topic comes from my experimental work (with colleagues) on the links between immune function, temperament, affective state, stress responsiveness and growth rate in sheep and cattle. I’ve reframed and re-written the paper in terms of these issues to make it more relevant to readers working in farm animal biology.

Reviewer 5 Report

11. ? such a “long term view”?? Certainly not held by many ecologists who have studied these things!

25 mutualism has long been what goes on see Odum  and many other early ec:ologists. I am not sure that it has to be “proved” because otherwise how would animals be sentient, make choices and decisions???

33: this definition of agency should be made clearer and repeated later and could be one of the main points of the paper.  

37-43 This is just a confirmation of what ecologists/ethologist have been saying for a long time.  The new material could be that relating to the immunological system which is interesting and should be emphasised. Suggest cutting of  long debates discussing various peoples rather curious views . Much too much jargon in the text.

As a simple ecologist/philosopher/ethologist,  much of this paper seems to be an unnecessary discussion of esoteric views which have little substance. There seems also to be a certain level of ignorance concerning how natural systems  work implicit in some of these debates.

54-57  Do we really need to study what all these might or might not mean related to the question? Your point is lost in the “theories”… and so is common sense which like it or not does rule much of our lives… of course the animal has to make decisions and be an agent, if he is sentient so why the long discussion?

See quote from Mary Midgeley ( who you should refer to she has been emphasising all of  this for a very long time)

page 168 in Science and Poetry 2002: “ The upper or softer layers are then ranked as relatively superficial because they do not give an ultimate explanation. They are considered amateurish, non-serious --- as Berkeley put it: the property of the vulgar, now called folk psychology”…

The great unwashed had some experience of animals and manage to teach them, work with them etc etc because of their “common sense/ folk psychology or folk knowledge” understanding for centuries…. The “upper hard layer”  ( so called “scientific” approach) and their instigators ( scientists and philosophers) are now usually quite ignorant about this common sense knowledge of the natural ecosystems as they have been city born and bred with no everyday contact and experience. Are they  now  discovering that the common sense approach has something to be said for it?!

67 the American Psychologist ( Gibson) should have done his reading!!

Why “affordance”, is it not just  that it is difficult for the cow but not for the goat because  they have different bodies /mind beings?

81 yes this is an important point that van Uexhull made and is now being made again and again in publications as it points to the importance of learning rather than fixed “instinct”.

83 paragraph omit.

90 There are much more simple examples in many different animals, see Avital and Jablonka for a book of references.

96-110  paragraph omit or at least simplify , cant see this tells us anything interesting about the question posed.

111 Quote OK except for all the curious terminology, perhaps rewrite so it is simply saying what you think it does say. I cant imagine that all readers of Animals will be totally convinced by its present form, or understand it.

118-120 OK but simplify again eg:  “The organism, according to Walsh, has  a dynamic mutualistic relationship to its environment rather than being controlled by genetic long term changes, as we all know”

omit 123-126.

omit 134 don't understand this  “operational sense” and is it worth it for the argument you are making?

138 put Jablonkas definition with Shannons to start with, say what YOU think information is and why you discuss  others if you like, but this reads as if it has been made purposely obscure rather than purposefully clear, which is what we want  if we are going to work together from different disciplines.

147-150 Spooky, do people really think like this???

151 This is the most interesting piece of the paper, but again don't complicate it, make it simple and avoid quoting names as well as giving them a reference.

206-210 omit or shorten without more terms.

reduce the next few paragraphs by half.

248 & rest, there has been enormous amount of work on hunger and thirst and how the physiology/neurophysiology/brain/behaviour inter-relates since the late 1960’s so why is this discussed at such length?

258-261 A point worth making.

276 et con, this is interesting but again needs serious cutting.

344 A relational “spectrum” ? Again a one sentence summary would do.

368 why is it either a “process ontology” or a “substance”? Cant it be both? if not why not.

373 Surely anyone who has any idea about animals knows that they can learn??  I cant see that pigs learning to identify a noise illustrates anything new at all… unless  it is believed that pigs are robots and not sentient… Animals are generally already able to “garner information from their environment” or they would not be animals with sensory systems, so they do not need selecting for it! Knowing this however,  might actually require that  you provide them with “information to garner”, something intensive husbandry is not good at.

I am sympathetic with your central message but putting it across supposing that the animal is some sort of robot, until proved otherwise is I feel not helpful to  understanding.

There is some interesting material on physiological and immune system mutualism that should be emphasised, and the simple more convincing conclusion of animal agency could be made clearer without digressions into many curious and not helpful beliefs.

I would recommend the paper for publishing BUT when it is considerably shortened and some rewriting.

The English is fine, but long winded and one has the impression that the author is trying to convince himself that there is some “mutualism” between the animal and the environment  and that he is saying something new, even though he knows that it is well known.

Author Response

Thank you for your comments. My responses are entered with red highlight

11. ? such a “long term view”?? Certainly not held by many ecologists who have studied these things!

The abstract is now revised so wording is changed substantially

25 mutualism has long been what goes on see Odum  and many other early ec:ologists. I am not sure that it has to be “proved” because otherwise how would animals be sentient, make choices and decisions???

The paper has been refocused substantially and mutualism is not now a focus. I am certainly not implying that mutualism had been proved. A new Section 6 now replaces the discussion on mutualism and agency in the early manuscript

33: this definition of agency should be made clearer and repeated later and could be one of the main points of the paper.  

Defining and exploring agency is no longer a strong focus of the paper.

37-43 This is just a confirmation of what ecologists/ethologist have been saying for a long time.  The new material could be that relating to the immunological system which is interesting and should be emphasised. Suggest cutting of  long debates discussing various peoples rather curious views . Much too much jargon in the text.

 The revised paper has moved away from a focus on mutualism, and no longer leads to the conclusions outlined in old lines 37-43. Jargon is reduced throughout and new terms introduced in a more explanatory style.

As a simple ecologist/philosopher/ethologist,  much of this paper seems to be an unnecessary discussion of esoteric views which have little substance. There seems also to be a certain level of ignorance concerning how natural systems  work implicit in some of these debates.

 The paper is refocussed on examining the environment in relation to breeding and management of farm animals for resilience and to improve welfare

54-57  Do we really need to study what all these might or might not mean related to the question? Your point is lost in the “theories”… and so is common sense which like it or not does rule much of our lives… of course the animal has to make decisions and be an agent, if he is sentient so why the long discussion?

The through-line of the paper now focusses more on understanding animal environment interactions in terms of predictive activities of the brain. Recent work by Peters, McEwen and Friston on stress and allostasis is used to tie together the outside-in approach of understanding senses and motor actions, and the inside-out approach to understanding physiological and immunological activities.

Yes, sentience, and the need to interact with the internal and external environments, is not in question. The processes of animal environment interaction are examined to help understand how behaviour, affect, physiological regulation, growth and immune function are interdependent.

See quote from Mary Midgeley ( who you should refer to she has been emphasising all of  this for a very long time)

page 168 in Science and Poetry 2002: “ The upper or softer layers are then ranked as relatively superficial because they do not give an ultimate explanation. They are considered amateurish, non-serious --- as Berkeley put it: the property of the vulgar, now called folk psychology”…

The great unwashed had some experience of animals and manage to teach them, work with them etc etc because of their “common sense/ folk psychology or folk knowledge” understanding for centuries…. The “upper hard layer”  ( so called “scientific” approach) and their instigators ( scientists and philosophers) are now usually quite ignorant about this common sense knowledge of the natural ecosystems as they have been city born and bred with no everyday contact and experience. Are they  now  discovering that the common sense approach has something to be said for it?!

Thanks for this reference. Addressing the important epistemological issues of concern to Midgeley is not within scope of the revised and shortened paper. My favourite example of folk wisdom in animal management practices that enhance affect (and appetite) in farm animals is the study of French shepherds by Villalba JJ, Provenza FD, Catanese F, Distel RA (2015) Understanding and manipulating diet choice in grazing animals. Animal Production Science 55, 261–271. doi:10.1071/AN14449, but this too is beyond the scope of the paper.

67 the American Psychologist ( Gibson) should have done his reading!!

Why “affordance”, is it not just  that it is difficult for the cow but not for the goat because  they have different bodies /mind beings?

 Less emphasis is placed on affordances in the revised paper. Yes (of course) goats and cattle have different bodies which are related to how they interact with the environment. The point here is that the concept of affordances probes in deeper terms the relationship between the animal and the aspects of the environment it can sense and respond to.  

81 yes this is an important point that van Uexhull made and is now being made again and again in publications as it points to the importance of learning rather than fixed “instinct”.

83 paragraph omit.

This has been deleted from the revised ms

90 There are much more simple examples in many different animals, see Avital and Jablonka for a book of references.

This issue is not included within the revised ms

96-110  paragraph omit or at least simplify , cant see this tells us anything interesting about the question posed.

 Brief description of Patten’s work is retained, but the paragraph has been simplified and the importance of the mutualism reduced in the revised paper.

111 Quote OK except for all the curious terminology, perhaps rewrite so it is simply saying what you think it does say. I cant imagine that all readers of Animals will be totally convinced by its present form, or understand it.

 This paragraph has been removed during shortening and focussing the paper

118-120 OK but simplify again eg:  “The organism, according to Walsh, has  a dynamic mutualistic relationship to its environment rather than being controlled by genetic long term changes, as we all know”

 This paragraph has been removed during shortening and re-focussing the paper

omit 123-126.

 Discussion of environmental information has been removed from the paper. This is not because examining biology through the perspective of Information theory is not important, but removing this part helps make the paper more straight forward and focussed

omit 134 don't understand this  “operational sense” and is it worth it for the argument you are making?

 As above

138 put Jablonkas definition with Shannons to start with, say what YOU think information is and why you discuss  others if you like, but this reads as if it has been made purposely obscure rather than purposefully clear, which is what we want  if we are going to work together from different disciplines.

 As above

147-150 Spooky, do people really think like this???

 The application of information theory to biology by scientists like Avery, Friston, Weiner, Jablonka and Lamb, and of course a very large number of others, has provided many insights. In this view, animals are indeed in the business of using information to reduce uncertainty. In the process of reducing uncertainty, the animal develops an ability to predict and act. In information theory an innate or learnt ability to predict and act is an internal model of the uncertainty in the environment. A nice example is provided by circadian clocks. Almost all organisms have circadian clocks. In Shannon’s information theory, information is the reduction of uncertainty, and uncertainty is also described as surprise (or surprisal). For an organism without an ability to predict the change from dark to light, the change has great uncertainty. An internal clock, that synchronises the activities (molecular, behavioural etc as relevant to the organism) with the light dark phase, reduces surprise (uncertainty) in the event. Uncertainty has been reduced by internalising information about the environment. The organism with an internal clock is a model of the light dark phases of the world it inhabits.  Although the narrative account might not be compelling, part of the power of an information theory approach to biology the ability to describe these details in mathematical terms (eg Ramstead, M.J.D.; Badcock, P.B.; Friston, K.J.  Answering Schrödinger's question: a free-energy formulation. Physics of life reviews 2017).

As an aside, Odum is very good on energetics but does not appear to approach his discussion of information from a basis in Information theory, or through an alternative theory that describes inadequacies of Information theory to justify an alternative approach.

The circadian clock model is useful also to illustrate the concept of “expectation” as used in the Bayesian brain model. The cyclical molecular activity that constitutes the circadian clock can be said to generate an expectation of the light dark cycle in the external world. However day length is not constant. Like any mechanical clock, unless errors are corrected the clock will soon run out of phase with the time it is meant to predict. So a circadian clock is only of value to an organism if there is a mechanism to detect and correct errors between the predictions of the internal clock and actual external light and dark cycle. The process of prediction, detection and correction to update the next cycle of prediction is what neuroscientists describe as analogous to approximate Bayesian inference (or in short form the Bayesian brain). The circadian clock example also illustrates that the organism does not need a nervous system to use a predictive model in its interactions with its environment.

So Cohen’s formulation (line 148) “… a living organism is a 'contrivance' for creating information, while paying its due to entropy”, is a very concise summary of the issue.

In the revised manuscript I no longer address the topic “what is environmental information”.

151 This is the most interesting piece of the paper, but again don't complicate it, make it simple and avoid quoting names as well as giving them a reference.

 The presentation of this section has been simplified.

206-210 omit or shorten without more terms.

 Terminology has been simplified

reduce the next few paragraphs by half.

Most of these paragraphs have been replaced with new text or deleted

248 & rest, there has been enormous amount of work on hunger and thirst and how the physiology/neurophysiology/brain/behaviour inter-relates since the late 1960’s so why is this discussed at such length?

This text has also been removed

258-261 A point worth making.

 A new section examines boundaries (or more accurately the lack thereof) between the animal and its various environments

276 et con, this is interesting but again needs serious cutting.

 The section on the immune system has been reduced by about half and is now better focussed on neuro-immune interactions. The operating principle of the immune system is no  longer addressed

344 A relational “spectrum” ? Again a one sentence summary would do.

“Spectrum” is no longer discussed and this section has been replaced

368 why is it either a “process ontology” or a “substance”? Cant it be both? if not why not.

 Dupre, the cited reference, explores the difference in detail. In part he writes:

“The ontology of things, following the revival of atomism

in the seventeenth century, has been the dominant metaphysics

for most of the history of modern science. It is closely

connected to a further position that underlies conceptions

of scientific explanation, mechanicism. For mechanicism, the

way to understand or explain a phenomenon is to identify

the various constituent things that interact to generate the

phenomenon. Arrangements of constituents with particular

functions constitute mechanisms. Mechanicism sees living systems

as composed of things arranged in a hierarchy of

mechanisms. This is a strictly bottom-up perspective, related

to, if generally distinguished from, the often criticized but still

widely endorsed methodological approach of reductionism.

Process ontologists generally reject both mechanicism and

reductionism, for they notice that what maintains the patterns

of stability in the sea of process is not only the behaviour of

the entities that compose the pattern, but also the network of

relations between the patterns and their surroundings [3].”

The distinction is important. For instance the decades long debate in human psychology over whether there are basic emotions or whether emotions are constructed ( as best exemplified by Lisa Barrett’s work) is at its core a debate between a substance ontology and process ontology of emotion. The different conception of emotion that is bound up in the differing ontological viewpoints has strong implications for the study of affect in farm animals, but the issue lies beyond the scope of the paper.

Ontology is no longer mentioned in the revised paper.

373 Surely anyone who has any idea about animals knows that they can learn??  I cant see that pigs learning to identify a noise illustrates anything new at all… unless  it is believed that pigs are robots and not sentient… Animals are generally already able to “garner information from their environment” or they would not be animals with sensory systems, so they do not need selecting for it! Knowing this however,  might actually require that  you provide them with “information to garner”, something intensive husbandry is not good at.

 The example is not included to illustrate the common sense fact that pigs can learn. This section is also reworded, and is used to illustrate that structuring the information environment provides pigs with an opportunity to control social conflicts and leads to improvement in affective state, which is in line with your suggestion of providing them with “information to garner”.

Farm animals do vary in their cognitive ability and domestication has (unintentionally) selected animals with an ability to use information in the constructed niche(s) of farming. For instance, cognitive ability varies with coping style. So intentionally or unintentionally we are continuing to modify the cognitive ability of farm animals through genetic section. This issue is not addressed in the paper. As you suggest, improving the environment is the starting point for improving welfare, not forcing animals through selection to fit a minimal environment.

I am sympathetic with your central message but putting it across supposing that the animal is some sort of robot, until proved otherwise is I feel not helpful to  understanding.

I am not meaning to portray animals as robots. The style of presentation has changed and I hope this impression is no longer conveyed. Instrumentality of the environment as exemplified in stress does not imply the animal is a robot. Stress is described in more detail to illustrate the balance between environmental instrumentality and animal agency. 

There is some interesting material on physiological and immune system mutualism that should be emphasised, and the simple more convincing conclusion of animal agency could be made clearer without digressions into many curious and not helpful beliefs.

There is a stronger emphasis now on physiology and the immune system. 

I would recommend the paper for publishing BUT when it is considerably shortened and some rewriting.

The paper has been extensively re-written and reduced in length from 433 to 295 lines of text.

The English is fine, but long winded and one has the impression that the author is trying to convince himself that there is some “mutualism” between the animal and the environment  and that he is saying something new, even though he knows that it is well known.

Round 2

Reviewer 1 Report

p.p1 {margin: 0.0px 0.0px 0.0px 0.0px; font: 11.0px 'Helvetica Neue'; color: #000000; -webkit-text-stroke: #000000} p.p2 {margin: 0.0px 0.0px 0.0px 0.0px; font: 11.0px 'Helvetica Neue'; color: #000000; -webkit-text-stroke: #000000; min-height: 12.0px} span.s1 {font-kerning: none}

The manuscript is improved but not sufficiently. This is still a difficult read, despite the effort that has clearly been made to revise. Notwithstanding, the author’s claim, the objective remains obscure and the take home messages are simply not there. For example, it has not been made clear how a our understanding of the links between behaviour, temperament, immune function and metabolism will be improved by a better description of the environment from the perspective of the animal. 

The majority of the substance of the first two paragraphs of my original review still apply to this version of the manuscript. 

Reviewer 3 Report

 A consideration of the relationship between animals  and their environment

As I said in my last report, this is an interesting paper drawing upon a wide range of research. I have now read it several times, and it has now ‘grown’ on me. So, my main advice is for the paper to provide more ‘signposts’ and for the author to make their presence more felt in leading the reader through the wide-ranging material. A much more informative title would also help!

My comments will focus on the issue of mutuality.

1.    The issue being addressed in this paper is the special context of farmed animals. But, quite understandably, it also needs to consider the situation where the animal has more ‘control’ over their circumstances, where, it is claimed, there is “greater mutuality” (247-8). But, for exampe, in factory farming, has mutuality been eliminated, or taken a different form?

2.    On Gibson and Uexküll. I agree that Gibson was probably unaware of this work, but to judge from his reaction, say, to Koffka, I think Gibson would have rejected, rightly or wrongly, his “subjectivism.” Gibson was doing his best to undermine subject-object dualism. The present article does not take on this issue, and I am not suggesting it  does. But there are practical implications. Within standard science, including veterinary science, the subjective, e.g. pain or boredom, is oftern dismissed as unreal!

3.    E.g. “The brain generates” (109). This ‘’centralization’ of the brain is difficult to square with the wider claim about animal-environment mutuality.

4.    A ‘sharper’ conclusion would be good. Why not end with specific examples of kinds of way to improve the welfare of animals in the light of the paper?

5.    This is just an historical note, but mutuality was a big issue in the early days of ‘animal psychology.’ As Lloyd Morgan insisted, an animal is no "mere puppet in the hand of circumstances" (Morgan, 1894, p. 338):

“The pendulum swing of opinion has, under the teaching of Professor Weismann, swung so far in the direction of the non-acceptance of the hereditary transmission of characters individually acquired through intelligent adjustment or otherwise; that the part played by consciousness in the evolution of the higher and more active animals is apt to pass unnoticed or unrecorded. It is well, therefore, to put in a reminder that a great number of animals would never reach the adult state in which they pass into the hands of the comparative anatomist save for the acquisition of experience, and the effective use of the consciousness to which they are heirs; that their survival is due, not only to their possession of certain structures and organs, but, every whit as much, to the practical use to which these possessions are put in the give and take of active life.” (Morgan, 1900, pp. 310-311).

Specific points

106: terms like ‘stimuli’ really are at odds with mutuality. that term orginally referred to a nasty object for prodding animas!

243-4: unpack/clarify. things are moving bit fast here!

Typos

22: (to) cope

57: identify(ing)

67: identify(ing)

68: (U)mwelt – I think it is normally spelt this way even in the English literature.

Reviewer 4 Report

The author has completely transformed the earlier version and now explains the approach and relevance well. Whilst some points would not be accepted by all, the paper now makes a useful contribution to understanding of the area.

Reviewer 5 Report

 Better that it was, but it stil is very wordy unnecessarily. I have made some details suggestions for cutting and rewording. Too many quotes of not helpful references!
